# Genetic basis of *Arabidopsis thaliana* responses to infection by naïve and adapted isolates of turnip mosaic virus

**Anamarija Butkovic[1†‡], Thomas James Ellis[2†], Ruben Gonzalez[1†‡], Benjamin Jaegle[2], Magnus Nordborg[2]\*, Santiago F Elena[1,3]\***

[1]Instituto de Biología Integrativa de Sistemas (I2SysBio), CSIC-Universitat de València, Parc Científic UV, València, Spain; [2]Gregor Mendel Institute (GMI), Austrian Academy of Sciences, Vienna BioCenter, Doktor-Bohr-Gasse, Vienna, Austria; [3]The Santa Fe Institute, Santa Fe, United States

**\*For correspondence:**
magnus.nordborg@gmi.oeaw.ac.at (MN);
santiago.elena@csic.es (SFE)

[†]These authors contributed equally to this work

**Present address:** [‡]Institut Pasteur, Paris, France

**Abstract** Plant viruses account for enormous agricultural losses worldwide, and the most effective way to combat them is to identify genetic material conferring plant resistance to these pathogens. Aiming to identify genetic associations with responses to infection, we screened a large panel of *Arabidopsis thaliana* natural inbred lines for four disease-related traits caused by infection by *A. thaliana*-naïve and -adapted isolates of the natural pathogen turnip mosaic virus (TuMV). We detected a strong, replicable association in a 1.5 Mb region on chromosome 2 with a 10-fold increase in relative risk of systemic necrosis. The region contains several plausible causal genes as well as abundant structural variation, including an insertion of a *Copia* transposon into a Toll/interleukin receptor (TIR-NBS-LRR) coding for a gene involved in defense, that could be either a driver or a consequence of the disease-resistance locus. When inoculated with TuMV, loss-of-function mutant plants of this gene exhibited different symptoms than wild-type plants. The direction and severity of symptom differences depended on the adaptation history of the virus. This increase in symptom severity was specific for infections with the adapted isolate. Necrosis-associated alleles are found worldwide, and their distribution is consistent with a trade-off between resistance during viral outbreaks and a cost of resistance otherwise, leading to negative frequency-dependent selection.

## eLife assessment

This manuscript presents **important** findings that inform the genetic underpinnings of the model plant *Arabidopsis'* resistance to turnip mosaic virus (TuMV). The strength of the evidence in the manuscript is **exceptional**, with very large sample sizes, careful controls, multiple follow-up experiments, and broadening to the evolutionary context. The evidence provides robust support for each of the manuscript's conclusions and could pave the way for functional studies.

## Introduction

Plant viruses represent an enormous threat to crop yields and food security (*Tomlinson, 1987*; *Oerke, 2006*; *Jones, 2021*). Infected plants are difficult to treat and cure, so it is of vital importance to identify genetic material that is resistant to infection (*Monnot et al., 2021*). In spite of this, the genetic architecture of plant responses to viral infections has received much less attention than the response to bacterial and fungal pathogens (*Bartoli and Roux, 2017*; *Monnot et al., 2021*). In agricultural settings, plants are predominantly grown as monocultures, which results in more virulent and specialized viruses that cause more detrimental effects on the host (*McDonald and Stukenbrock, 2016*;

*González et al., 2019*). Through their specialization in one host species, viruses also evolve better counter defenses than their naïve counterparts (*Brosseau et al., 2020*).

In this study, we investigated the response of the model plant *Arabidopsis thaliana* (L.) HEYNH to its pathogen TuMV (TuMV; species *Turnip mosaic virus*, genus *Potyvirus*, family *Potyviridae*). Potyviruses affect a wide variety of crops, especially from the families *Brassicaceae* and *Solanaceae*, and are among the most widespread crop pathogens worldwide (*Revers and García, 2015*). TuMV is among the most damaging of the potyviruses (*Tomlinson, 1987*), and also has a high incidence in wild populations of *A. thaliana* (*Pagán et al., 2010*). The extensive genetic resources available in *A. thaliana* make it a useful system for investigating mechanisms of viral resistance in plants. In addition, *A. thaliana* belongs to the same family (*Brassicaceae*) as agriculturally important crops such as cauliflower, cabbage, turnip, or rapeseed; thus, a large array of relevant crop viruses can infect and be studied in *A. thaliana* (*Pagán et al., 2010*; *Ouibrahim and Caranta, 2013*). Viral outbreaks are frequent in natural populations of *A. thaliana*, and there is substantial genetic variation in resistance, indicating that viral coevolution represents a meaningful selection pressure in this species (*Pagán et al., 2010*).

Several previous genome-wide association (GWA) studies of the response of *A. thaliana* to viral infection have been carried out. *Rubio et al., 2019* used TuMV and 317 lines grown under field conditions, while *Butković et al., 2021* also used TuMV but 450 lines were kept in laboratory conditions. *Montes et al., 2021* and *Liu et al., 2022* used 156 and 496 inbred lines, respectively, to measure the response of *A. thaliana* to infection by cucumber mosaic virus under controlled conditions. *Hoffmann et al., 2023* used 100 inbred lines to study the response of *A. thaliana* to infection by cauliflower mosaic virus. These studies have demonstrated that genetic variation for virus response exists, and that individual loci with large effects on virus response segregate in *A. thaliana* populations.

Here, we report the results of GWA studies using two isolates of TuMV and one of the largest sets of *A. thaliana* (1050) inbred lines studied so far. We compare an 'ancestral' isolate of TuMV that was obtained from a calla lily (*Zantedeschia* sp.) and was naïve to *A. thaliana* (*Chen et al., 2003*), to its 'evolved' descendant that had been submitted to 12 serial passages of experimental evolution on *A. thaliana* Col-0 line (*González et al., 2019*). The two virus isolates differ in two non-synonymous mutations fixed during their adaptation to Col-0 and in their infection phenotypes. The evolved virus has mutations in amino acids T1293I (cylindrical inclusion protein; CI) and N2039H (viral genome-linked protein; VPg) (*Navarro et al., 2022*). CI is involved in viral genome replication and cell-to-cell movement (*Deng et al., 2015*), and has been shown to interact with the photosystem I PSI-K protein in *A. thaliana* (*Jiménez et al., 2006*). VPg protein is involved in replication and intracellular movement (*Wu et al., 2018*), and is a hub for interactions with other viral proteins (*Bosque et al., 2014*; *Martínez et al., 2023*). Mutations in VPg have been pervasively observed in evolution experiments of potyviruses in *A. thaliana*, in all cases resulting in increased infectivity, virulence and viral load (*Agudelo-Romero et al., 2008*; *Hillung et al., 2014*; *González et al., 2021*; *Navarro et al., 2022*; *Ambrós et al., 2024*; *Melero et al., 2023*). We aim to identify plant genes that play a role in TuMV infection and whether some of these genes may respond differentially to each viral isolate.

## Results
### Stronger disease phenotypes in response to the evolved TuMV isolate

We inoculated 1050 lines of *A. thaliana* (*Alonso-Blanco et al., 2016*) with the ancestral (*Chen et al., 2003*) or evolved (*González et al., 2019*) isolates of TuMV and characterized the response to each one based on four phenotypes: (*i*) infectivity (proportion of plants showing symptoms), (*ii*) AUDPS (area under the disease progress stairs), a measure of disease progress (*Simko and Piepho, 2012*), (*iii*) severity of symptoms on a semi-quantitative scale from 0 to 5 (*Figure 1A*; *Butković et al., 2020*), and (*iv*) necrosis (a binary trait reflecting presence/absence of systemic necrosis, or stage 5 of the severity-of-symptoms scale). We did not measure viral accumulation, but note this is significantly correlated with the intensity of symptoms within the Col-0 line (Corrêa et al. 2020), although it is not clear if this correlation occurs in all lines. Furthermore, throughout the manuscript, we refer to necrosis as the most severe stage of infection: systemic necrosis that results in the death of the plant (*Figure 1A*) rather than to the hypersensitive response that induces local necrotic spots and limits the spread of a pathogen. As a negative control, we mock-inoculated 2100 plants in the same trays as the

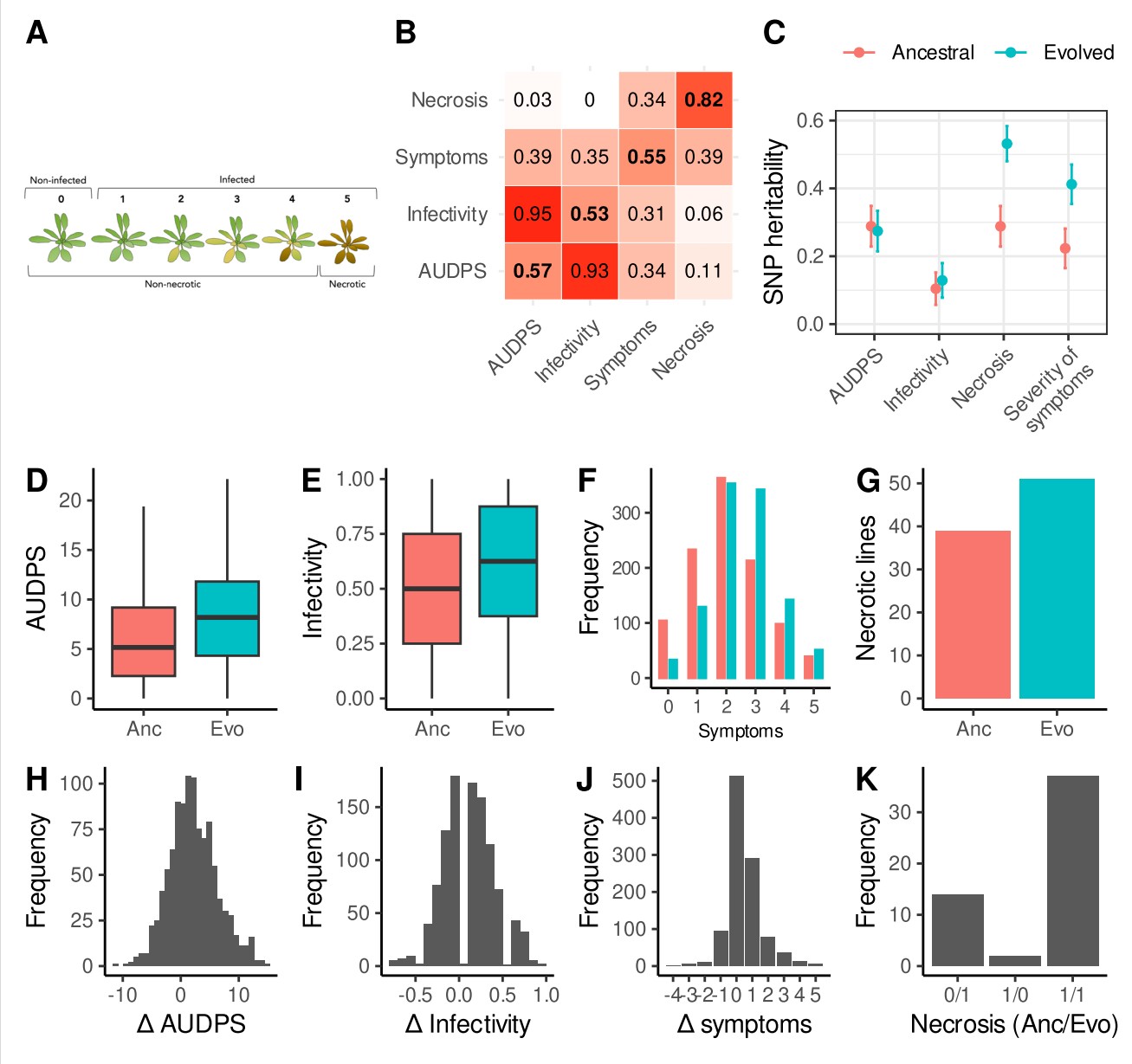

**Figure 1.** Disease phenotypes. (**A**) Illustration of the scale used to evaluate the severity of symptoms: (0) no symptoms or healthy plant (1) mild symptoms without chlorosis, (2) visible chlorosis, (3) advanced chlorosis, (4) strong chlorosis and incipient necrosis, (5) necrosis and death of the plant. (**B**) Correlation matrix between disease phenotypes in response to the ancestral (upper left) and evolved (lower right) isolates. The diagonal shows correlations between the same phenotype in response to each viral isolate. (**C**) SNP heritability for each trait in response to each virus. (**D–G**) Disease phenotypes across lines in response to the ancestral (red) and evolved (blue) turnip mosaic virus (TuMV) isolates. (**H–J**) Differences in disease phenotypes in response to each viral isolate (evolved minus ancestral). (**K**) Number of lines showing necrosis in response to the ancestral or/and evolved isolates; 1 indicates the presence of necrosis. For clarity, the 997 lines showing no necrosis for either isolate (0/0) are not shown.

treated plants and did not observe disease symptoms in any case, indicating that there was no cross-contamination between plants.

Patterns of correlations between traits were very similar between the two viral isolates (off-diagonal elements of the correlation matrix between phenotypes were largely symmetric), with strong correlations between AUDPS and infectivity, and weak correlations between necrosis and other traits (*Figure 1B*). Traits showed low to moderate SNP heritability, with a stronger correlation between genotype and severity of symptoms and necrosis in response to the evolved virus (*Figure 1C*). This shows that phenotyping was repeatable, and the disease traits measured show genetic variation.

Lines infected with the evolved isolate showed more severe disease symptoms than those infected with the ancestral one (*Figure 1D–G*). We assessed phenotype differences using linear models that account for cohort effects, and calculated *p*-values by permutation. On average, we found a 37% increase in AUDPS (*p*<0.001), a 27% increase in infectivity (*p*<0.001), and a 23% increase in severity of symptoms (*p*<0.001). We also found a 31% increase in necrosis, although this is not statistically significant when cohort effects are taken into account (*p*=0.08). On average, the history of adaptation to Col-0 is associated with increased virulence across the panel of *A. thaliana* inbred lines studied here.

Despite this overall trend, there was substantial variation in the direction of effects between lines. Although necrosis was highly repeatable between viral isolates, we found only moderate correlations for AUDPS, infectivity, and severity of symptoms in response to the two isolates (*Figure 1B*, diagonal elements). When comparing the responses of *A. thaliana* lines to the evolved isolate relative to the ancestral one, 53.21% of lines exhibited increased infectivity, 23.08% showed the same level of infection, and 23.70% were less infected by the evolved virus. In terms of disease progression (AUDPS values), the evolved viral strain performed better in 64.49% of the lines, the same in 9.25%, and worse than the ancestral strain in 26.25%. Two lines exhibited no necrosis in response to the evolved isolate, despite previously showing necrosis when exposed to the ancestral virus. Conversely, 14 lines that did not display necrosis for the ancestral virus exhibited necrosis when infected by the evolved isolate (*Figure 1I–L*). Adaptation of TuMV to Col-0 thus tends to enhance the virus efficacy in other lines but does not guarantee increased infectivity across the range of *A. thaliana* lines, pointing to a complex interaction between viral isolates and plant genotypes.

## A strong genetic association with necrosis and severity of symptoms on chromosome 2

We used the multi-trait GWA analysis implemented in the software package LIMIX to identify individual genetic loci that correlate with the response to each TuMV isolate (*Lippert et al., 2014*). LIMIX assesses the response to each viral isolate jointly, and identifies loci associated with (*i*) a shared response to both isolates and (*ii*) specific responses to individual viral isolates.

We found a strong peak of association between the common response to both isolates via both severity of symptoms and necrosis and a 1.5 Mb region on chromosome 2 (*Figure 2A–B*). Lines with the less frequent (henceforth 'minor') allele at the most strongly associated SNP showed 23.2- and 11.9-fold increase in necrosis, as well as a 1.83- and 1.75-fold increase in severity of symptoms in response to the ancestral and evolved isolates, respectively. The most strongly associated SNP in this region is found inside locus *AT2G14080*, encoding a Toll/Interleukin-1 Receptor leucine-rich-repeat (TIR-LRR-NBS) disease resistance gene (*Supplementary file 2a*; *Meyers et al., 2003*). The region of association also covers three additional plausible candidate genes (*Supplementary file 2a*), including a strong association with *DRP3B*, which has been previously linked to viral replication (*Wu et al., 2018*), and weaker associations with *ALD1* (AGD-like defense protein) and *AT2G14070* (involved in wound repair). The marked association with disease phenotypes and the presence of multiple candidate genes are good evidence that this region plays a causative role in the response to viral infection.

We used two approaches to confirm the robustness of this association. First, we repeated the experimental procedure using 51 lines that showed systemic necrosis in the initial experiment and 65 that did not test the hypothesis that the observed association is a coincidence due to the small number of necrotic lines. We recovered the association in the same region on chromosome 2, with a peak at position 5,927,469 (*Figure 2—figure supplement 1*). Second, we compared disease symptoms in the Col-0 genotype to those in T-DNA mutants for two candidate loci: *AT2G14080* and *DRP3B* (*Supplementary file 2a*; *Figure 2F*). Consistent with the evolved isolate being adapted to Col-0, there was a significant increase in the severity of symptoms in Col-0 in response to the evolved isolate (*U*=15.5, *p*=0.007). Mutant *at2g14080* showed significantly increased severity of symptoms relative to Col-0 in response to the ancestral isolate (*U*=9, *p*=0.001), but no significant difference from Col-0 in response to the evolved one (*U*=49, *p*=0.971), suggesting an antiviral role for this gene that has been overcome by the evolved isolate. In contrast, *drp3b* plants showed a significant difference from Col-0 in response to the evolved isolate (*U*=20.5, *p*=0.023), but not the ancestral isolate (*U*=47.5, *p*=0.853), suggesting a potential proviral role for this gene gained by the evolved viral isolate. The association between the region on chromosome 2 and disease phenotypes is readily repeatable, and two candidate genes in the region show a significant difference in the severity of symptoms from Col-0.

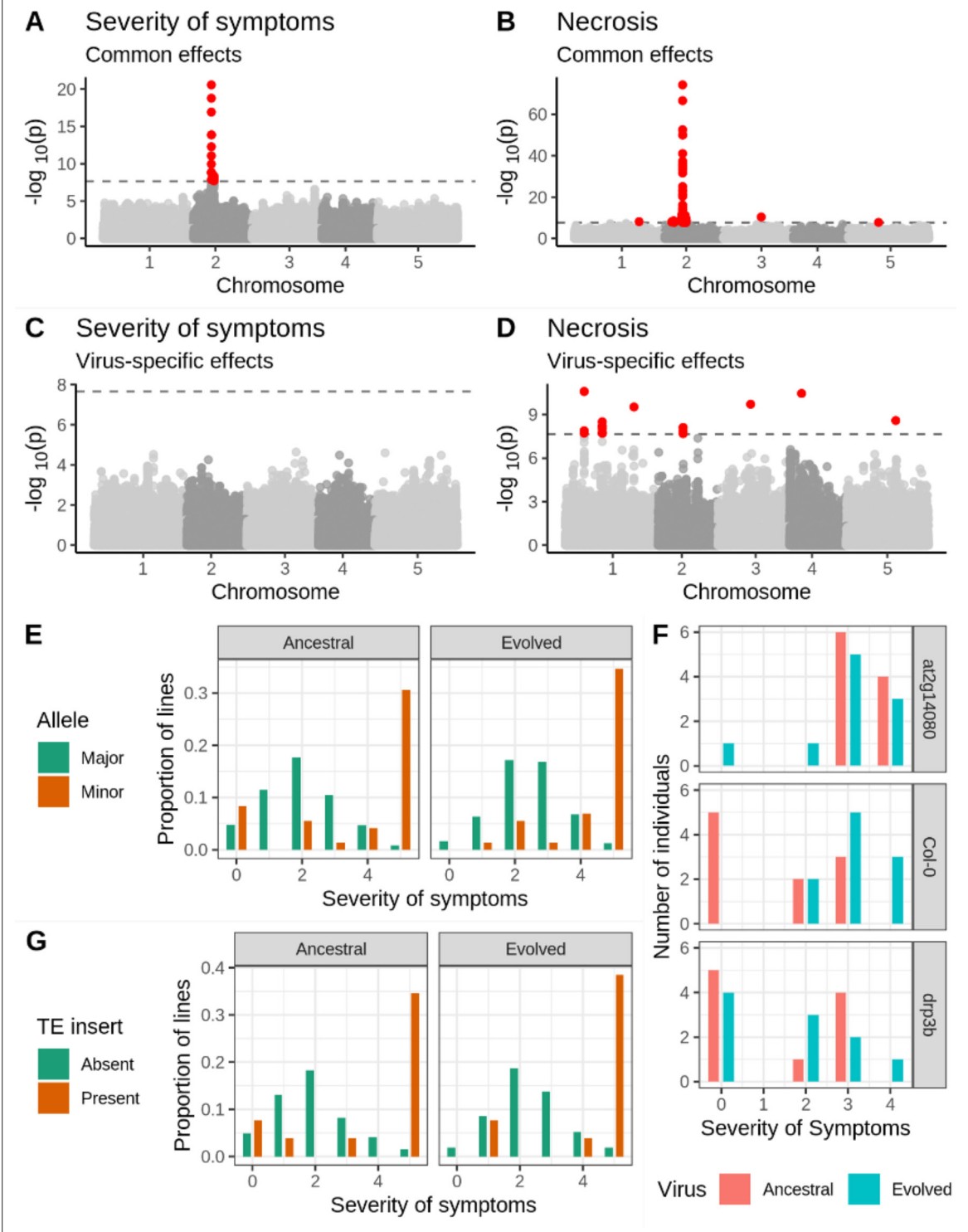

**Figure 2.** Genetic associations with severity of symptoms and systemic necrosis. (**A, B**) associations with a common response to both viral isolates. (**C, D**) associations with isolate-specific responses. SNPs in red indicate markers with −log₁₀p-values above the Bonferroni-corrected significance threshold. (**E**) Severity of symptoms in response to each viral isolate for accessions with the major and minor alleles at the most strongly associated SNP. (**F**) Severity of symptoms in response to each virus for plants of T-DNA knockout mutants for two major candidate genes and Col-0 wild-type controls. (**G**) Severity of symptoms in response to each virus for lines with and without a TE insertion in *AT2G14080*.

The online version of this article includes the following figure supplement(s) for figure 2:

**Figure supplement 1.** Manhattan plots for the association between SNPs and necrosis in the replicate dataset of 118 lines.

*Figure 2 continued on next page*

*Figure 2 continued*

**Figure supplement 2.** Manhattan plots for associations with severity and symptoms and necrosis in a model conditioned on the most strongly associated SNP (5,927,469 in chromosome 2).

**Figure supplement 3.** Linkage disequilibrium between SNPs associated with necrosis.

**Figure supplement 4.** Manhattan plots for common and virus-specific associations with area under the disease progress stairs (AUDPS) and infectivity.

## Additional associations with necrosis

In addition to the strong association with severity of symptoms and necrosis, we identified 13 loci associated with necrosis (*Figure 2*; *Supplementary file 2a*). Six of these associations were common responses to both viral isolates and seven with isolate-specific responses. These loci include the annotated disease resistance genes *RECOGNITION OF PERONOSPORA PARASITICA 7* (*RPP7*; encoding an LRR and NB-ARC domains-containing protein), *BAK1-INTERACTING RECEPTOR-LIKE KINASE 2* (*BIR2*), and *AT1G61100*, which are strong candidates for an involvement in the response to TuMV. However, minor-allele frequencies at all but one of these loci are below 5%, and the presence of the strong association on chromosome 2 may cause false-positive associations at other loci. We, therefore, repeated the GWA analysis, including the genotype at the most strongly associated SNP as a cofactor (*Segura et al., 2012*). This revealed two loci with a significant association with the common response to both viruses, and eight loci with a virus-specific response (*Figure 2—figure supplement 2* and *Supplementary file 2b*). All but one (chromosome 4 position 273,465) of the isolate-specific associations overlapped between the analyses with and without the major association on chromosome 2. Moreover, linkage disequilibrium between these loci is weak (*Figure 2—figure supplement 3*), suggesting that associations are not due to chance long-range correlations. Notably, *AT1G61100* encodes a TIR-class disease resistance gene and is detected both with and without the major association as a cofactor.

We did not identify any significant associations with AUDPS or infectivity (*Figure 2—figure supplement 4*), consistent with the low heritability for these traits (<30%; *Figure 1C*).

## A *Copia* element insertion in the primary candidate gene

We explored haplotypes in the region associated with necrosis in 157 genomes assembled de novo from long-read PacBio sequencing data as part of a different study (https://1001genomes.org/). We aligned the sequences in the region around the strong association for necrosis and severity of symptoms. To assess patterns of synteny, we looked for homology with the two most strongly associated genes (*AT2G14080* and *DRP3B*) and the eight annotated genes or transposable elements from the TAIR10 genome annotation on either side of these genes using BLAST. This analysis revealed many structural polymorphisms (*Figure 3*), including abundant intergenic indel variation, duplications, and a large presence/absence polymorphism downstream of *AT2G14080*, containing gene *CYTOCHROME P450 FAMILY 705 SUBFAMILY A POLYPEPTIDE 13* (*CYP705A13*) and a block of transposable elements. At least one line (9470) shows a large-scale inversion for the entire region. The region around the strong association for necrosis and severity of symptoms appears to be a hotspot for structural variation.

We next looked for structural polymorphism that might be causal for the disease phenotypes. Thirteen inbred lines harbored a *Copia*-element inside the first intron of *AT2G14080*. We tried to genotype the presence/absence of this element in the full sample using short-read sequencing data, but this turned out to be unachievable due to the repetitiveness of the sequence (see Materials and methods). Among those lines for which PacBio genomes are available, there was a marked increase in severity of symptoms for lines with the TE insert (*Figure 2G*). Nine and ten lines showed necrosis in response to the ancestral and evolved viral isolates respectively, corresponding to a 23- and 21-fold increase in risk of necrosis for lines with this insertion. This element remains a promising candidate for a causal polymorphism.

## Alleles associated with susceptibility are geographically overdispersed

The minor allele at the SNP most strongly with increased disease symptoms, is spread throughout the global range of the GWA panel, from Siberia to North America (*Figure 4A*). This observation is curious, because we would expect selection to limit the spread of deleterious mutations. To investigate

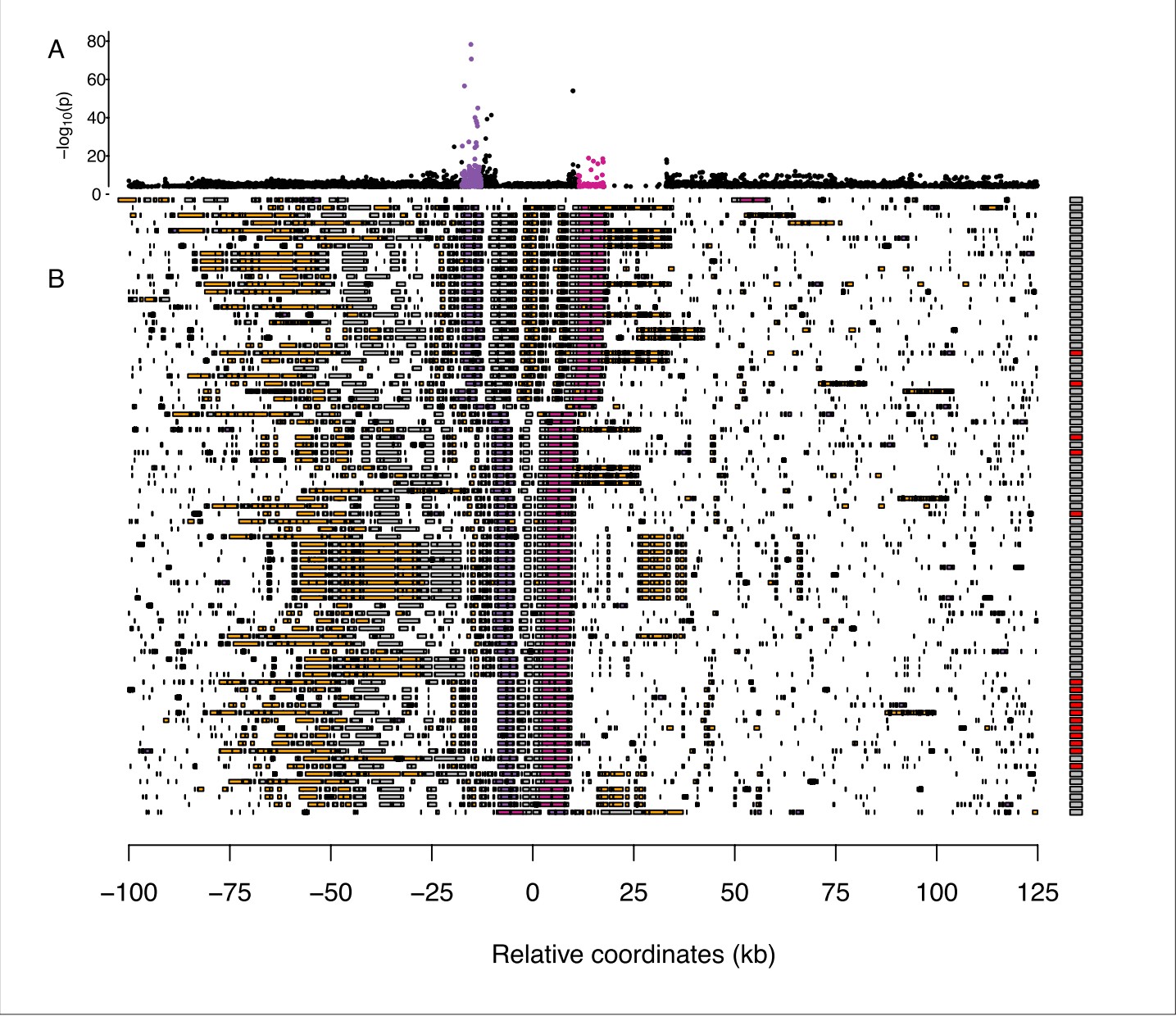

**Figure 3.** Multiple sequence alignment around the major association with severity of symptoms and necrosis on chromosome 2. (**A**) An enlarged view of the peak of association with necrosis in *Figure 2B*. (**B**) A summary of structural variation in assembled genomes. Transposable elements are shown in orange and coding genes in gray, with candidate genes *AT2G14080* and *DRP3B* highlighted in purple and pink, respectively. Boxes on the right indicate necrosis (red) in response to either virus or no necrosis (gray). For clarity, only half of the non-necrotic lines are plotted.

whether this broad spatial distribution could be due to chance, we quantified the mean geographic distance between lines with the minor allele at this SNP, as well as each of the most strongly associated SNPs in the other peaks identified by GWA for necrosis that were significant after accounting for the major association as a cofactor (*Supplementary file 2b*, *Figure 2D*, and *Figure 2—figure supplement 2*). We then compared these distances to the mean distances between lines with the minor allele at 10,000 randomly chosen loci with similar minor allele counts. For most associated loci, distances between lines with susceptible alleles are well within the distribution of randomly chosen loci, consistent with genetic drift allowing minor alleles to reach modest frequency on a local spatial scale. In contrast, lines harboring the minor alleles at the strong association on chromosome 2 are further apart on average than 98.5% of randomly chosen alleles at similar frequencies (*Figure 4B*). This allele is found at a similar frequency in all but three admixture groups identified by reference

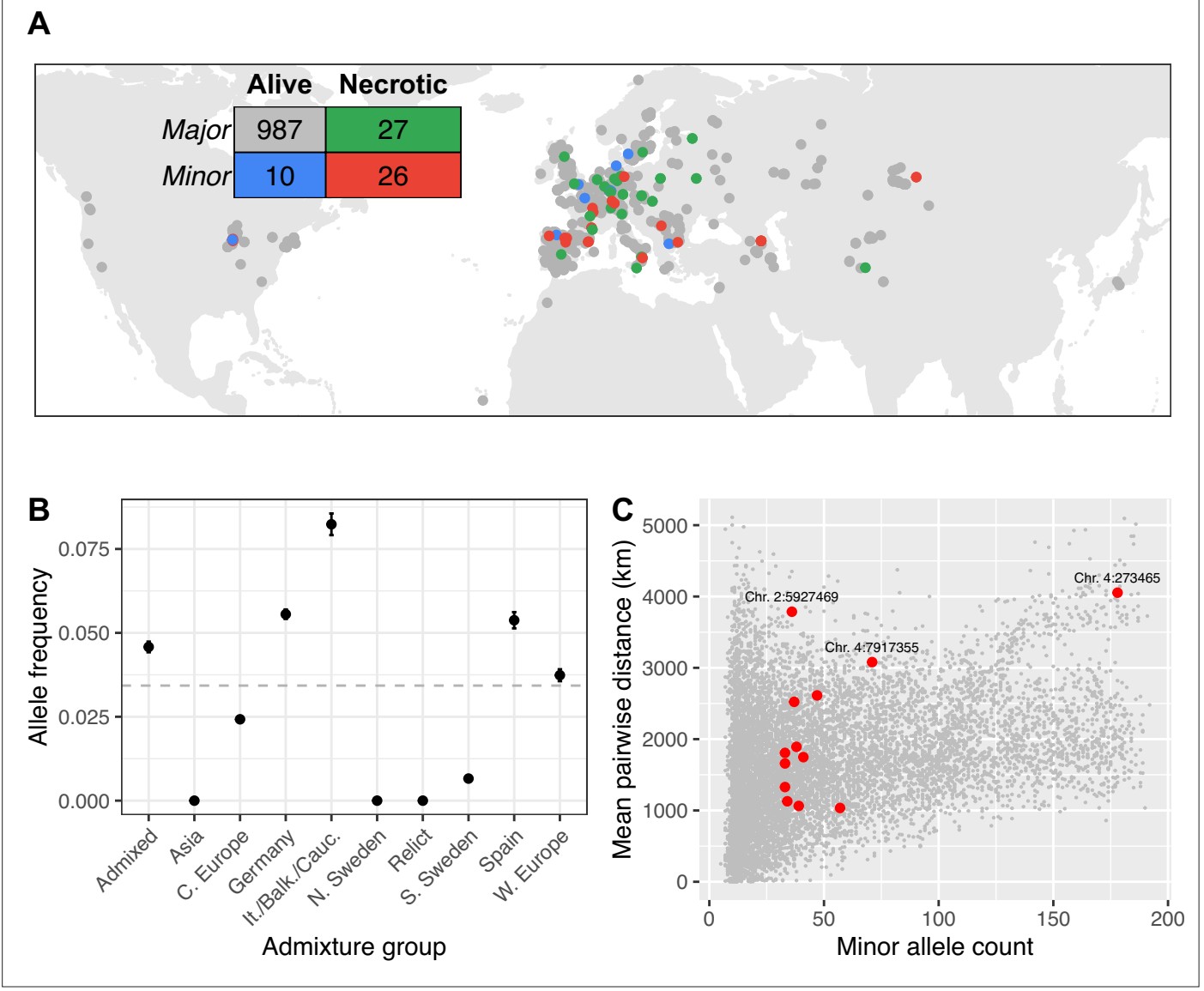

**Figure 4.** Global distribution of necrotic alleles. (**A**) Global distribution of genotypes (minor/major alleles at the SNP most strongly associated with systemic necrosis), and phenotypes (whether lines showed systemic necrosis or not). The inset table shows the numbers of each genotype/phenotype; colors correspond to those on the map. (**B**) Frequencies of minor alleles at chromosome 2 position 5,927,469 within each admixture group. The horizontal line indicates global frequency. (**C**) Mean distances between pairs of loci harboring the minor allele at the 13 associated loci (red) and between minor alleles at 10,000 randomly chosen loci.

(**Alonso-Blanco et al., 2016**), so this overdispersion cannot be accounted for by an association with a particular ancestry with a high dispersal rate (**Figure 4C**). We found similar spatial overdispersion for the associated loci on chromosome 4 at positions 273,465 and 7,917,355. Alleles associated with systemic necrosis at the major association seem to be spatially overdispersed, in a way that is independent of genetic background, and that is unlikely to be due to genetic drift alone.

## Discussion

We report results from the largest GWA study done so far aiming to identify *A. thaliana* genes involved in the infection response to one of its natural pathogens, TuMV, using an ancestral isolate naïve to *A. thaliana*, and a second isolate that had been experimentally evolved on *A. thaliana* line Col-0. The comparison of the results for the two isolates has pinpointed candidate host targets for virus adaptation.

## Natural variation of disease-related traits in response to different TuMV isolates

The majority of inbred lines infected with the evolved viral lineage showed more severe disease symptoms compared to plants infected with the ancestral isolate (*Figure 1D-G*). Our study confirms that the adaptation to one susceptible *A. thaliana* line, in this case, Col-0 enables potyviruses to perform better in otherwise poorly susceptible lines. This observation was first made by *Lalić et al., 2010* after evolving a tobacco etch virus isolate from tobacco in the susceptible line L*er*-0. Upon inoculation of 18 other lines, they observed that the evolved isolate exhibited greater infectivity compared to the ancestral one. However, using a much larger panel of lines, we also found substantial variation in both the magnitude and direction of the response to the two viral lineages between the inbred lines, showing that adaptation to one genotype results in a complex response across host genotypes. Subsequent plant-viral co-evolution is thus likely to depend on the range of host genotypes the virus encounters and, hence, the extent to which it is able to specialize.

## The genetic architecture of TuMV resistance

We identified a 1.5 Mb region on chromosome 2 that was strongly associated with the severity of symptoms and necrosis. This region encompasses several genes (*Supplementary file 2a*), two of which are especially strong candidates for causing these phenotypes. First, *AT2G14080* encodes a known TIR-NBS-LRR family protein. These proteins recognize and block translation of viral transcripts in the cytoplasm, induce a hypersensitive response leading to cell death, act as a second line of defense if the initial antiviral response fails to restrict virus accumulation, and are involved in signaling pathways (*Li et al., 2001*; *Meyers et al., 2003*; *Bhattacharjee et al., 2009*; *Kobayashi et al., 2010*; *Marone et al., 2013*; *Van de Weyer et al., 2019*). TIR-NBS-LRR genes are the most numerous disease-resistance genes in the *A. thaliana* genome and are under diversifying selection to respond to as broad a spectrum of pathogens as possible (*Ellis et al., 2000*). This gene appears to have a significant role in the development of systemic necrosis during infection with both TuMV strains. Second, *DRP3B* encodes a self-assembling GTPase involved in fission and fusion of membranes in mitochondria and peroxisomes (*Fujimoto et al., 2009*; *Wu et al., 2018*). Its analog, *DRP2B*, has been shown to be co-opted by TuMV to enhance virulence, and treatment with dynamin-specific inhibitors suppresses growth by interfering with VPg (*Wu et al., 2018*). As previously noted, one of the two amino-acid differences between the ancestral and evolved TuMV isolates is in VPg (*Navarro et al., 2022*). These observations make *AT2G14080* and *DRP3B* strong candidate genes for a role in viral resistance, either independently or in tandem.

To test this possibility, we inoculated the ancestral and evolved isolates into KO mutants *at2g14080* and *drp3b*. We observed that loss of function of *AT2G14080* led to increased disease symptoms in response to the ancestral virus, but not the evolved virus (*Figure 2F*). This suggests that *AT2G14080* is efficient for the antiviral defense against the ancestral but not for the evolved isolate. The evolved viral isolate may be able to overcome the response imposed by *AT2G14080* due to the mutations acquired during its adaptation to Col-0, most probably because changes in the VPg viral protein allow it to evade detection by the TIR-NBS-LRR protein encoded by *AT2G14080*. In contrast, loss of function of *DRP3B* decreased symptoms relative to those in Col-0 in response to the evolved, but not the ancestral virus. This points to a possible proviral effect of the dynamin *DRP3B*. Potyviruses are able to recruit *DRP2B*, a homolog of *DRP3B*, for the viral replication complex (*Wu et al., 2018*). One explanation for the reduction in symptoms in *drp3b* plants is that the evolved isolate acquired the ability to co-opt *DRP3B* in a similar way to *DRP2B*, in a way that the ancestral isolate does not. Whether these two loci act independently or in an epistatic manner should be considered as a possibility, considering the complex interplay between viral genetic factors and environmental cues in determining the phenotypic outcomes of viral infections (*Bomblies et al., 2007*; *Lalić and Elena, 2013*).

Thirteen additional loci (besides the major association on chromosome 2) were significantly associated with the common or specific responses to the two viral isolates (*Figure 2—figure supplement 2*, *Supplementary file 2b*). The low linkage disequilibrium between these loci indicates that they segregate independently of the major association on chromosome 2, and of each other (*Figure 2—figure supplement 3*). However, minor alleles are also at very low global frequencies (*Supplementary file 2b*). On the one hand, this is consistent with the expectation that selection against such deleterious alleles keeps susceptible alleles rare. On the other hand, there is a substantial risk that alleles at such

low frequency should be associated with a binary phenotype at a similar frequency just by chance. As such, caution is warranted in interpreting a causative role for these associations without further evidence.

We found very little overlap with associations found in two previous GWA studies of TuMV resistance in *A. thaliana*. First, *Butković et al., 2021* identified genome-wide associations for AUDPS, severity of symptoms, and infectivity under very similar experimental conditions to those used here, but using two viral isolates that differed in their host range (one evolved as a specialist in more permissive *A. thaliana* lines and the other evolved as a generalist able to successfully infect more resistant lines) and fewer host lines. That analysis recovered the association with *AT2G14080* for both isolates, and identified eleven additional loci associated with the response to one or both viruses. These eleven loci identified were not recovered in our larger dataset. This lack of consistency can either indicate that these associations were spurious artifacts of a limited sample size or, alternatively, a true biological effect due to the different TuMV isolates used by *Butković et al., 2021* and in this study. Second, we did not detect any of the genes associated with viral load or infectivity in 317 lines grown under field conditions identified by *Rubio et al., 2019*, and nor did that study detect the association on chromosome 2 reported here. In this case, the lack of overlap may be explained not only by the differences in sample sizes, but also by the differences in traits measured, viral isolate, and environmental conditions.

## The *Copia*-element insertion is a candidate causative variant

We identified a *Copia*-element polymorphism in the first intron of *AT2G14080* which is strongly associated with severity of symptoms, and especially necrosis. Given this strong association, and the fact that first introns often harbor regulatory elements affecting transcription (*Gallegos and Rose, 2017*), this insertion is a strong candidate for a causative mutation. We note also that it need not be the only causal mutation. Whether it turns out to be casual or not, the abundance of structural variation in this region highlights the need for restraint when trying to identify any causal polymorphisms from SNP data alone, and more generally the limitation of reference-biased polymorphism data.

It is also interesting to consider whether structural variation is driven by selection for diversification of resistance genes, or whether particular structural variants themselves cause variation in disease resistance. It has been previously observed that plant disease-resistance genes often form clusters of duplicated copies within the genome, which would contribute to structural variation between genomes (*Leister, 2004*). In particular, TIR-NBS-LRR genes, of which *AT2G14080* is an example, are known to have undergone expansion by numerous large- and small-scale duplication and translocation events (*Leister, 2004*). However, we did not observe widespread duplications of *AT2G14080* or its domains, nor evidence of synteny with other regions of the genome that would indicate translocations, so in this case, a simple transposon insertion remains the most likely explanation.

## The distribution of susceptible alleles is consistent with frequency-dependent selection

Given that potyvirus outbreaks are common in nature (*Pagán et al., 2010*) and susceptibility to symptomatic infection can be deleterious, it is curious that alleles associated with increase disease symptoms should be at sufficiently high frequency to be detectable. There are three possible explanations for this. First, selection may be sufficiently weak or the virus geographically restricted, such that alleles can arise in one location and increase in frequency by genetic drift. If this were true, we would expect to see a clustered geographic distribution of susceptible alleles, reflected in under dispersion of distances between pairs of susceptible lines. While this is true for most of the loci with small effects on necrosis, susceptible alleles at the major association on chromosome 2 are spatially over-dispersed worldwide, and found throughout unrelated lines (*Figure 4*). These patterns are thus difficult to explain via genetic drift. Second, it may be that the genomic instability in the region surrounding this association leads to a high turnover of new structural mutations that decrease viral resistance. This is implausible because this would not generate a linkage with individual SNPs that could be detected by GWA. Moreover, we found a striking concordance between a single *Copia*-element insertion into *AT2G14080* and necrosis, suggesting that only a single variant is responsible for increased necrosis in this region. Neither genetic drift nor mutation-selection balance can thus explain the persistence of susceptible alleles at the major association for necrosis.

An alternative explanation is that susceptible alleles are beneficial in the absence of the virus. This would cause a fitness trade-off between alleles in this region, as has been previously demonstrated for many other pathogen-resistance systems in plants (*Bergelson et al., 2001*; *Todesco et al., 2010*). TuMV outbreaks are common in natural *A. thaliana* populations, but ephemeral (*Pagán et al., 2010*), causing susceptible alleles to decrease in frequency during outbreaks, but to increase at other times. This would lead to negative frequency-dependent selection maintaining the susceptible allele at a frequency in proportion to the frequency of TuMV outbreaks. Despite being at low frequency worldwide, the susceptible allele is found across the geographic and genealogical distribution of *A. thaliana*, indicating that the polymorphism must either be very old and has been maintained by balancing selection over long periods, or that alleles have spread by adaptive introgression during viral outbreaks. Our data do not allow us to distinguish between these hypotheses, nor are they mutually exclusive, but both are consistent with a fitness trade-off at this locus. Our results indicate that the worldwide geographic distribution of susceptible alleles is, therefore, consistent with negative frequency-dependent selection maintaining the susceptible allele at low, but non-zero, frequency.

## Materials and methods

### Viruses and inoculation procedure

We obtained the ancestral virus from infected *Nicotiana benthamiana* DOMIN plants inoculated with a transcript product from a p35STunos infectious plasmid that contains TuMV genome cDNA (GenBank line AF530055.2), corresponding to YC5 isolate from calla lily (*Chen et al., 2003*). This cDNA was under the control of the cauliflower mosaic virus 35 S promoter and a NOS terminator. We obtained the evolved virus via 12 passages of experimental evolution on *A. thaliana* line Col-0 (*González et al., 2019*). In both cases, we pooled symptomatic tissues, froze them with liquid $N_2$, and homogenized them into a fine powder using a Mixer Mill MM400 (Retsch GmbH, Haan, Germany). For inoculations, we mixed 100 mg of powdered liquid-nitrogen-frozen infected tissue in 1 mL of phosphate buffer (50 mM phosphate pH 7.0, 3% PEG6000, 10% Carborundum) and rubbed 5 µL of this mixture into three central rosette leaves per plant. To minimize the noise that significant variations in vegetative development could produce, we infected all the lines when they reached growth stage 3.2–3.5 in the *Boyes et al., 2001* scale, approximately 21 days after germination.

### Disease phenotypes in 1050 accessions

We screened 1050 lines from the *Arabidopsis* 1001 Genomes Project (*Alonso-Blanco et al., 2016*) for disease phenotypes (*Supplementary file 1*). We grew selected lines in climatic chambers under a long-day regime (16 hr light/8 hr darkness) with 24 °C day and 20 °C night, 45% relative humidity, and light intensity of 125 µmol $m^{-2}$ $s^{-1}$ using a 1:3 mixture of 450 nm blue and 670 nm purple LEDs. Due to limited growth-chamber space, we inoculated and phenotyped lines in four independent cohorts of 4800 plants each. Five lines in the fourth cohort that did not reach the proper size on the day of the inoculation were inoculated four days after the other lines. We inoculated eight plants per line with the ancestral virus and another eight with the evolved virus, as well as two mock-inoculated plants that served as negative controls. We placed four lines inoculated with each viral isolate and the corresponding mock-inoculated plants in the same tray and randomized tray positions every day.

We measured four disease-related traits daily for 21 days post inoculation (dpi), at which point infection phenotypes reached a steady plateau. We measured the proportion of infected plants ('infectivity'), AUDPS (a measure of disease progression using the distribution of infected plants over the 21 dpi; *Simko and Piepho, 2012*), severity of symptoms (based on the scale shown in *Figure 1A*), and necrosis (systemic necrosis or the most severe stage of disease). Necrosis was considered present in a line if at least one plant showed full-leaf systemic necrosis on all leaves, as shown with number 5 in *Figure 1A*.

To test for statistically significant differences in disease phenotypes in response to the two viruses we constructed linear models using the phenotype as the response variable, and viral isolate and cohort as explanatory variables. For infectivity and necrosis, we used a Binomial generalized linear model (GLM) with a logit link function. For AUDPS, we log-transformed the data to improve normality, and fitted a GLM with a Gaussian link function. For severity of symptoms, we applied a cumulative link model with a logit link function using the R package 'ordinal' (*Christensen RHB, 2022*). Because

p-value estimation is sensitive to departures from normality on the scale of the link function, we calculated a null distribution for the difference due to viral isolate by permutation. For each phenotype, we randomly shuffled the response variable 1000 times and refitted the linear model. The (one-sided) p-value is then the proportion of permuted samples with a greater difference in phenotype between the two viral isolates than was found for the observed data. Since it was previously found that the evolved virus induced a stronger response in the reference line Col-0 (*Butković et al., 2020*), we report one-sided *p*-values.

## Genetic associations

We used the Python package LIMIX 3.0.3 (*Lippert et al., 2014*) in Python 3.7.3 for multi-trait GWA analyses of disease-related traits based on the statistical framework from *Korte et al., 2012* and Segura. LIMIX assesses the response to each virus jointly, and identifies loci associated with (*i*) a shared response to both viruses, and (*ii*) specific responses to individual viral isolates.

We looked for associations with the 10,709,949 SNPs published by *Alonso-Blanco et al., 2016*. We used a liberal minor-allele-frequency cut-off of 3% to allow us to detect rare variants associated with necrosis, since only 3.7% and 4.9% of lines showed necrosis in response to the ancestral and evolved TuMV isolates respectively, leaving 2,257,599 SNPs. We accounted for cohort effects by including these as fixed cofactors, and for population structure by modeling the covariance between lines due to relatedness as a random effect (*Bergelson et al., 2001*; *Kang et al., 2008*). In a second analysis, we repeated this GWA for necrosis including the genotype at chromosome 2 position 5,927,469 as a covariate to account for the confounding effects of this locus (*Korte et al., 2012*). For all GWA, we used a significance threshold of $-\log P = 7.65$, corresponding to a Bonferroni-corrected p-value of 0.05.

We estimated narrow-sense SNP heritability using the R package 'Sommer' 4.1.4 (*Covarrubias-Pazaran and Zhang, 2016*) by regressing phenotypes onto the matrix of pairwise relatedness between individuals (*Yang et al., 2010*). To estimate the variance explained by the major association for necrosis, and to assess the sensitivity of GWA and heritability to the presence of this locus we also repeated both GWA and heritability estimates including genotypes at the SNP most strongly associated with necrosis as a covariate.

To identify plausible candidate genes, we first identified all coding genes within the region spanned by SNPs with $-\log_{10}$ p-values above the Bonferroni-corrected significance threshold and recorded their annotated functions from arabidopsis.org. We highlighted genes as candidate genes if they had functions plausibly related to disease response or defense.

## Further investigation of the major genetic association

### Replication cohort

To confirm that the association between necrosis and symptom severity with the candidate region on chromosome 2 (see results) is not an artifact of the small number of lines showing necrosis, we tested an additional cohort including all 51 lines that had previously shown necrosis in response to one or both viruses together with 67 randomly chosen non-necrotic lines from the previous screen. This confirmation cohort was inoculated and grown in the same way as the four previous cohorts.

### Knock-out lines

We assessed disease resistance for knock-out (KO) mutants for two candidate genes within the region associated with necrosis and severity of symptoms. *AT2G14080* encodes for a TIR-NBS-LRR protein, and *AT2G14120* for the DRP3B protein. We ordered T-DNA knock-out (KO) mutants from the NASC stock center (http://arabidopsis.info/BrowsePage). We chose mutants that (*i*) were to be in the Col-0 background, (*ii*) had T-DNA inserts that cause gene knock-outs, (*iii*) were homozygous, and (*iv*) had already been related to virus infection and selected SALK homozygous line SALK_105230 C (NASC ID: N656836) to study *AT2G14080* and SALK_112233 C (N667049) for *DRP3B*. We grew mutant and wild-type plants as previously described with 10 replicates per mutant/control, as well as two replicates of mock-inoculated controls. We visually inspected inoculated plants daily for 21 dpi noting the number of infected plants and the severity of their symptoms. We assess the statistical significance of differences between genotypes or viral treatment with Mann-Whitney *U* tests.

## Structural variation around locus *AT2G14080*

### Confirming the insertion using long-read genome assemblies

We examined haplotype structure in the 225 Kb surrounding the region associated with necrosis using data from 161 full genomes sequenced using PacBio long reads (*Arabidopsis* 1001 Genomes Consortium), and assembled using the methods previously described for the 1001 Genomes Plus Project (*Jaegle et al., 2023*). We extracted sequences around the peak of association, close to *AT2G14080*. We identified the precise location of that gene, as well as eight annotated genes and transposable elements on each side from the Araport 11 database, using BLAST (*Camacho et al., 2009*) if they showed 70% identity and were at least 70% of the length of the gene or transposable element in the reference genome. When comparing whole genomes, it is not meaningful to plot SNP positions with the coordinate system of a reference genome, so to visualize the results we plotted haplotypes on a relative scale centered at the midpoint between *AT2G14080* and *AT2G14120* (*DRP3B*), since these genes were largely syntenic between lines and flanked most of the region associated with necrosis. Finally, we sorted lines based on the distance between those two genes (*Figure 4*).

To confirm the insertions observed in the necrotic/susceptible lines 351, 870, 7346, and 9102, we mapped the raw PacBio reads to the new assembly using Minimap2 (*Li and Alkan, 2021*) with the option '-ax map-pb' and checked if any breaks coincided with the insertion position that would suggest an artifact of the assembly. This was not the case.

### Genotyping the insertion in the 1001 Genomes

We attempted to confirm the presence of the insertion in the full sample using existing short-read sequencing data (*Alonso-Blanco et al., 2016*). We used multiple approaches, all based on the flanking sequence of the insertion. First, using two different mappers, bwa-mem (*Li and Durbin, 2009*) and bowtie (*Langmead et al., 2009*), we mapped all the short-read data to the insertion sequence plus 5 Kb of flanking sequence (on either side) from line 870, filtering the bam files using samtools 'rmdup' (*Bonfield et al., 2021*) to remove duplicates. For each line, the number of paired reads mapping to each side of the insertion borders were extracted and counted. We assumed that the presence of such reads would confirm the presence of the insertion, whereas lines without the insertion would only have paired reads flanking the whole insertion. However, the mapping quality at the flanking regions was too low for this approach to work. Many flanking regions had no coverage from short-read sequencing (even when duplicated reads were kept). The reasons for this are not clear, but presumably reflect high levels of repetitiveness and high levels of polymorphism in the region. In a second approach, we use bbmap (*Bushnell et al., 2017*) to look for the exact sequence at the insertion site (±50 bp) within the raw fastq file. However, when we used the four lines with a confirmed insertion from de novo long-read assemblies as a control, this approach only worked in two cases. Again, the reasons are not clear, but the approach is clearly too unreliable for genotyping.

## Geographic distribution of major and minor necrosis alleles

We plotted the worldwide distribution of necrosis phenotypes and genotypes at the most strongly associated SNP using R packages 'maps' and 'mapdata' (*Becker et al., 2018*; *Becker et al., 2021*). To test whether the lines harboring minor alleles are more or less geographically dispersed than would be expected by chance, we identified 10,000 loci with similar minor allele counts (between 33 and 178) to the SNPs associated with necrosis (*Supplementary file 2b*), and the lines that harbor those alleles. We then compared the mean distance between lines harboring the SNP associated with necrosis to the distribution of distances between lines with the randomly chosen alleles.

## Acknowledgements

We thank Paula Agudo, Francisca de la Iglesia, and Joanna Gunis for their excellent technical support. Work was supported by grant PID2019-103998GB-I00 funded by MCIN/AEI/10.13039/501100011033 and Generalitat Valenciana grants GRISOLIAP/2018/005 and PROMETEO/2019/012 to SFE. RG were supported by contract BES-2016–077078 funded by MCIN/AEI/10.13039/501100011033 and 'ESF investing in your future.' MN was funded by ERC AdG 789037 - EPICLINES, ERA-CAPS (FWF I 3684-B25), and by the GMI.

## Additional information

### Competing interests
Magnus Nordborg: Reviewing editor, *eLife*. The other authors declare that no competing interests exist.

### Funding

| Funder | Grant reference number | Author |
|---|---|---|
| Ministerio de Ciencia e Innovación | PID2019-103998GB-I00 | Santiago F Elena |
| Generalitat Valenciana | PROMETEO/2019/012 | Santiago F Elena |
| Ministerio de Ciencia e Innovación | BES-2016-077078 | Ruben Gonzalez |
| European Research Council | ERC AdG 789037 - EPICLINES | Magnus Nordborg |
| Osterreichischer Wissenschaftsfonds | FWF I 3684-B25 | Magnus Nordborg |

The funders had no role in study design, data collection and interpretation, or the decision to submit the work for publication.

### Author contributions
Anamarija Butkovic, Ruben Gonzalez, Data curation, Formal analysis, Validation, Investigation, Visualization, Methodology, Writing – original draft, Writing – review and editing; Thomas James Ellis, Data curation, Software, Formal analysis, Investigation, Visualization, Methodology, Writing – original draft, Writing – review and editing; Benjamin Jaegle, Software, Formal analysis, Investigation, Methodology; Magnus Nordborg, Conceptualization, Supervision, Funding acquisition, Writing – original draft; Santiago F Elena, Conceptualization, Supervision, Funding acquisition, Writing – original draft, Project administration, Writing – review and editing

### Author ORCIDs
Thomas James Ellis ⬤ http://orcid.org/0000-0002-8511-0254
Magnus Nordborg ⬤ http://orcid.org/0000-0001-7178-9748
Santiago F Elena ⬤ http://orcid.org/0000-0001-8249-5593

Reviewer #1 (Public Review): https://doi.org/10.7554/eLife.89749.3.sa1
Reviewer #2 (Public Review): https://doi.org/10.7554/eLife.89749.3.sa2
Reviewer #3 (Public Review): https://doi.org/10.7554/eLife.89749.3.sa3
Author Response https://doi.org/10.7554/eLife.89749.3.sa4

## Additional files

### Supplementary files
• Supplementary file 1. Full list of *A. thaliana* lines for the GWAS analysis. (EXCEL).

• Supplementary file 2. Significant SNPs in the GWA analysis of the ancestral and evolved TuMV isolates and posterior conditional GWA analysis on the significant SNP on the chromosome 2.

• MDAR checklist

### Data availability
Phenotype data and code to reproduce the analyses presented here are given at GitHub (copy archived at *Ellis, 2023*) and Zenodo. Sequence data are available from the 1001 Genomes Project website (*Alonso-Blanco et al., 2016*). Unless otherwise stated analyses and plotting were done in R 4.0.3 (*R Development Core Team, 2020*) under RStudio Server Pro 1.3.1093-1 (*RStudio Team, 2020*).

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
