## [Editor Report · eLife assessment]

This manuscript presents **important** findings that inform the genetic underpinnings of the model plant *Arabidopsis'* resistance to turnip mosaic virus (TuMV). The strength of the evidence in the manuscript is **exceptional**, with very large sample sizes, careful controls, multiple follow-up experiments, and broadening to the evolutionary context. The evidence provides robust support for each of the manuscript's conclusions and could pave the way for functional studies.

---

## [Referee Report · Reviewer #1 (Public Review)]

In this manuscript, Butkovic et al. perform a genome-wide association (GWA) study on *Arabidopsis thaliana* inoculated with the natural pathogen turnip mosaic virus (TuMV) in laboratory conditions, with the aim to identify genetic associations with virus infection-related parameters. For this purpose, they use a large panel of *A. thaliana* inbred lines and two strains of TuMV, one naïve and one pre-adapted through experimental evolution. A strong association is found between a region in chromosome 2 (1.5 Mb) and the risk of systemic necrosis upon viral infection, although the causative gene remains to be pinpointed.

This project is a remarkable tour de force, but the conclusions that can be reached from the results obtained are unfortunately underwhelming. Some aspects of the work could be clarified, and presentation modified, to help the reader.

---

## [Referee Report · Reviewer #2 (Public Review)]

The manuscript presents a valuable investigation of genetic associations related to plant resistance against the turnip mosaic virus (TuMV) using *Arabidopsis thaliana* as a model. The study infects over 1,000 *A. thaliana* inbred lines with both ancestral and evolved TuMV and assesses four disease-related traits: infectivity, disease progress, symptom severity, and necrosis. The findings reveal that plants infected with the evolved TuMV strain generally exhibited more severe disease symptoms than those infected with the ancestral strain. However, there was considerable variation among plant lines, highlighting the complexity of plant-virus interactions.

A major genetic locus on chromosome 2 was identified, strongly associated with symptom severity and necrosis. This region contained several candidate genes involved in plant defense against viruses. The study also identified additional genetic loci associated with necrosis, some common to both viral isolates and others specific to individual isolates. Structural variations, including transposable element insertions, were observed in the genomic region linked to disease traits.

Surprisingly, the minor allele associated with increased disease symptoms was geographically widespread among the studied plant lines, contrary to typical expectations of natural selection limiting the spread of deleterious alleles. Overall, this research provides valuable insights into the genetic basis of plant responses to TuMV, highlighting the complexity of these interactions and suggesting potential avenues for improving crop resilience against viral infections.

Overall, the manuscript is well-written, and the data are generally high-quality. The study is generally well-executed and contributes to our understanding of plant-virus interactions.

---

## [Referee Report · Reviewer #3 (Public Review)]

Summary of Work

This paper conducts the largest GWAS study of *A. thaliana* in response to a viral infection. The paper identifies a 1.5 MB region in the chromosome associated with disease, including SNPs, structural variation, and transposon insertions. Studies further validate the association experimentally with a separate experimental infection procedure with several lines and specific T-DNA mutants. Finally, the paper presents a geographic analysis of the minor disease allele and the major association. The major take-home message of the paper is that structural variants and not only SNPs are important changes associated with disease susceptibility. The manuscript also makes a strong case for negative frequency-dependent selection maintaining a disease susceptibility locus at low frequency.

Strengths and Weaknesses

A major strength of this manuscript is the large sample sizes, careful experimental design, and rigor in the follow-up experiments. For instance, mentioning non-infected controls and using methods to determine if geographic locus associations were due to chance. The strong result of a GWAS-detected locus is impressive given the complex interaction between plant genotypes and strains noted in the results. In addition to the follow-up experiments, the geographic analysis added important context and broadened the scope of the study beyond typical lab-based GWAS studies. I find very few weaknesses in this manuscript.

Support of Conclusions

The support for the conclusions is exceptional. This is due to the massive amount of evidence for each statement and also due to the careful consideration of alternative explanations for the data.

Significance of Work

This manuscript will be of great significance in plant disease research, both for its findings and its experimental approach. The study has very important implications for genetic associations with disease beyond plants.

---

## [Author Response]

The following is the authors’ response to the original reviews.

**Reviewer #1 (Public Review):**
In this manuscript, Butkovic et al. perform a genome-wide association (GWA) study on *Arabidopsis thaliana* inoculated with the natural pathogen turnip mosaic virus (TuMV) in laboratory conditions, with the aim to identify genetic associations with virus infection-related parameters. For this purpose, they use a large panel of *A. thaliana* inbred lines and two strains of TuMV, one naïve and one pre-adapted through experimental evolution. A strong association is found between a region in chromosome 2 (1.5 Mb) and the risk of systemic necrosis upon viral infection, although the causative gene remains to be pinpointed.This project is a remarkable tour de force, but the conclusions that can be reached from the results obtained are unfortunately underwhelming. Some aspects of the work could be clarified, and presentation modified, to help the reader.
**(Recommendations For The Authors):**
It is important to note that viral accumulation and symptom development do not necessarily correlate, and that only the former is a proxy for "virus performance". These concepts need to be clear throughout the text, so as not to mislead the reader.

This has been explained better in line 118-120, “Virus performance has been removed.

Sadly, only indirect measures of the viral infection (symptoms) are used, and not viral accumulation. It is important to note that viral accumulation and symptom development do not necessarily correlate and that only the former is a proxy for "virus performance". These concepts need to be clear throughout the text, so as not to mislead the reader. The mention of "virus performance" in line 143 is therefore not appropriate, nor is the reference to viral replication and movement in the Discussion section.

"Virus performance" was removed. Also, the reference to viral replication and movement in the Discussion section has been removed.

Now we mention: “We did not measure viral accumulation, but note this is significantly correlated with intensity of symptoms within the Col-0 line (Corrêa et al. 2020), although it is not clear if this correlation occurs in all lines.”

Since symptoms are at the center of the screen, images representing the different scores in the arbitrary scales should ideally be shown.

Different Arabidopsis lines would look different and this could mislead a reader not familiar with the lines. In order to make a representation of our criteria to stablish the symptoms, we believe that a schematic representation is clearer to interpret. Here are some pictures of different lines showing variating symptoms:

**Author response image 1. sa4fig1:** 

Statistical analyses could be added to the figures, to ease interpretation of the data presented.

Statistical analysis can be found in methods. We prefer to keep the figure legend as short as possible.

The authors could include a table with the summary of the phenotypes measured in the panel of screened lines (mean values, range across the panel, heritability, etc.).

These data are plotted in Fig. 1. We believe that repeating this information in tabular form would not contribute to the main message of the work. Phenotype data and the code to reproduce figure 1 are available at GitHub (as stated in Data Availability), anyone interested can freely explore the phenotypes of the screened lines.

The definition of the association peak found in chromosome 2 could be explained further: is the whole region (1.5 Mb) in linkage disequilibrium? How many genes are found within this interval, and how were the five strong candidates the authors mention in line 161 selected? It is also not clear which are these 5 candidates, apart from AT2G14080 and DRP3B - and among those in Table 1 (which, by the way, is cited only in the Discussion and not in the Results section)? Why were AT2G14080 and DRP3B in particular chosen?

We have replaced Table 1 with an updated Table S1 listing all genes found within the range of significant SNPs for each peak. We now highlight a subset of these genes as candidate genes if they have functions related to disease resistance or defence, and mentioned them explicitly in the text (lines 173-179). We have explicitly described how this table was constructed in the methods (lines 525-538).

Concerning the validation of the association found in chromosome 2 (line 169 and onward): the two approaches followed cannot be considered independent validations; wouldn't using independent accessions, or an independent population (generated by the cross between two parental lines, showing contrasting phenotypes, for example) have been more convincing?

We aim to compare the hypothesis that the association is due to a causal locus to the null hypothesis that the observed association is a fluke due to, for example, the small number of lines showing necrosis. If this null hypothesis is true then we would not expect to see the association if we run the experiment again using the same lines. An alternative hypothesis is that the genotype at the QTL and disease phenotypes are not directly causally linked, but are both correlated with some other factor, such as another QTL, or maternal effects. We agree that an independent sample would be required to exclude the latter hypothesis, but argue that the former is the more pertinent. We have edited the text to be explicit about the hypothesis we are testing, and altered the language to shift the focus from ‘validation’ to ‘confirming the robustness’ of the association (line 182).

Regarding the identification of the transposon element in the genomic region of AT2G14080: is the complementation of the knock-out mutant with the two alleles (presence/absence of the transposon) possible to confirm its potential role in the observed phenotype?

This could be feasible but we cannot do it as none of the researchers can continue this project.

On the comparison between naïve and evolved viral strains: is the evolved TuMV more virulent in those accessions closer to Col-0?

This is not something we have looked at but would certainly be an interesting follow-up investigation.

The Copia-element polymorphism is identified in an intron; the potential functional consequences of this insertion could be discussed. In the example the authors provide, the transposable element is inserted into the protein-coding sequence instead.

We now state explicitly that such insertions are expected to influence expression; beyond that we can only speculate. We have removed the reference to the insertion in the coding sequence.

The authors state in line 398 that "susceptibility is unquestionably deleterious" - is this really the case? Are the authors considering susceptibility as the capacity to be infected, or to develop symptoms? Viral infections in nature are frequently asymptomatic, and plant viruses can confer tolerance to other stresses.

We have tone down the expression and clarify our wording: “Given that potyvirus outbreaks are common in nature (Pagán et al., 2010) and susceptibility to symptomatic infection can be deleterious”

Additional minor comments:In Table 1, Wu et al., 2018 should refer to DRP2A and 2B, not 3B.

We have removed Table 1 altogether.

Line 126: a 23% increase in symptom severity is mentioned, but how is this calculated, considering that severity is measured in four different categories?

This is the change in mean severity of symptoms between the two categories.

Figure 1F: "...symptoms"

Fixed.

Line 179: "...suggesting an antiviral role..."

Changed.

Lines 288-300: This paragraph does not fit into the narrative and could be omitted.

It has been removed and some of the info moved to the last paragraph of the Intro, when the two TuMV variants were presented.

Lines 335-337: The rationale here is unclear since DRP2B will also be in the background - wouldn't DRPB2B and 3B be functionally redundant in the viral infection?

Our results suggest that DRPB3B is redundant with DRPB2B for the ancestral virus but not for the evolved viral strain. We speculate that the evolved viral isolate may have acquired the capacity to recruit DRPB3B for its replication and hence it produces less symptoms when the plant protein is missing.

We have spotted a mistake that may have add to the confusion. Originally the text said “In contrast, loss of function of DRP3B decreased symptoms relative to those in Col-0 in response to the ancestral, but not the evolved virus”. The correct statement is “In contrast, loss of function of DRP3B decreased symptoms relative to those in Col-0 in response to the evolved, but not the ancestral virus.”

**Reviewer #2 (Public Review):**
The manuscript presents a valuable investigation of genetic associations related to plant resistance against the turnip mosaic virus (TuMV) using *Arabidopsis thaliana* as a model. The study infects over 1,000 *A. thaliana* inbred lines with both ancestral and evolved TuMV and assesses four disease-related traits: infectivity, disease progress, symptom severity, and necrosis. The findings reveal that plants infected with the evolved TuMV strain generally exhibited more severe disease symptoms than those infected with the ancestral strain. However, there was considerable variation among plant lines, highlighting the complexity of plant-virus interactions.A major genetic locus on chromosome 2 was identified, strongly associated with symptom severity and necrosis. This region contained several candidate genes involved in plant defense against viruses. The study also identified additional genetic loci associated with necrosis, some common to both viral isolates and others specific to individual isolates. Structural variations, including transposable element insertions, were observed in the genomic region linked to disease traits.Surprisingly, the minor allele associated with increased disease symptoms was geographically widespread among the studied plant lines, contrary to typical expectations of natural selection limiting the spread of deleterious alleles. Overall, this research provides valuable insights into the genetic basis of plant responses to TuMV, highlighting the complexity of these interactions and suggesting potential avenues for improving crop resilience against viral infections.Overall, the manuscript is well-written, and the data are generally high-quality. The study is generally well-executed and contributes to our understanding of plant-virus interactions. I suggest that the authors consider the following points in future versions of this manuscript:1. Major allele and minor allele definition: When these two concepts are mentioned in the figure, there is no clear definition of the two words in the text. Especially for major alleles, there is no clear definition in the whole text. It is recommended that the author further elaborate on these two concepts so that readers can more easily understand the text and figures.

We agree that the distinction between major/minor alleles and major/minor associations in our previous manuscript may have been confusing. In the current manuscript we now define the minor allele at a locus as the less-common allele in the population (line 167). We have removed references to major/minor associations, and instead refer to strong/weak associations.

2. Possible confusion caused by three words (Major focus / Major association and major allele): Because there is no explanation of the major allele in the text, it may cause readers to be confused with these two places in the text when trying to interpret the meaning of major allele: major locus (line 149)/ the major association with disease phenotypes (line 183).

See our response to the previous comment.

3. Discussion: The authors could provide a more detailed discussion of how the research findings might inform crop protection strategies or breeding programs.

We would prefer to restrain speculating about future applications in breeding programs.

**(Recommendations For The Authors):**
1. Stacked bar chart for the Fig 1F. It is recommended that the author use the form of a stacked bar chart to display the results of Fig 1F. On the one hand, it can fit in with the format of Fig 1D/E/G, on the other hand, it can also display the content more clearly.

We think the results are easier to interpret without the stacked bar chart.

2. Language Clarity: While there are no apparent spelling errors, some sentences could be rewritten for greater clarity, especially when explaining the results in Figure 1 and Figure 2.

We have reviewed these sections and attempted to improve clarity where that seemed appropriate.

There are some possibilities to explore in the future. For example: clarity of mechanisms for the future. While the study identifies genetic associations, it lacks an in-depth exploration of the underlying molecular mechanisms. Elaborating on the mechanistic aspects would enhance the scientific rigor and practical applicability of the findings.

Yes, digging into the molecular mechanisms is an ongoing task and will be published elsewhere. It was out of the scope of this already dense manuscript.

**Reviewer #3 (Public Review):**
Summary of WorkThis paper conducts the largest GWAS study of A . thaliana in response to a viral infection. The paper identifies a 1.5 MB region in the chromosome associated with disease, including SNPs, structural variation, and transposon insertions. Studies further validate the association experimentally with a separate experimental infection procedure with several lines and specific T-DNA mutants. Finally, the paper presents a geographic analysis of the minor disease allele and the major association. The major take-home message of the paper is that structural variants and not only SNPs are important changes associated with disease susceptibility. The manuscript also makes a strong case for negative frequency-dependent selection maintaining a disease susceptibility locus at low frequency.Strengths and WeaknessesA major strength of this manuscript is the large sample sizes, careful experimental design, and rigor in the follow-up experiments. For instance, mentioning non-infected controls and using methods to determine if geographic locus associations were due to chance. The strong result of a GWAS-detected locus is impressive given the complex interaction between plant genotypes and strains noted in the results. In addition to the follow-up experiments, the geographic analysis added important context and broadened the scope of the study beyond typical lab-based GWAS studies. I find very few weaknesses in this manuscript.Support of ConclusionsThe support for the conclusions is exceptional. This is due to the massive amount of evidence for each statement and also due to the careful consideration of alternative explanations for the data.Significance of WorkThis manuscript will be of great significance in plant disease research, both for its findings and its experimental approach. The study has very important implications for genetic associations with disease beyond plants.
**(Recommendations For The Authors):**
Line 41 - Rephrase, not clear "being the magnitude and sign of the difference dependent on the degree of adaptation of the viral isolate to *A. thaliana*."

Now it reads: “When inoculated with TuMV, loss-of-function mutant plants of this gene exhibited different symptoms than wild-type plants, where the scale of the difference and the direction of change between the symptomatology of mutant and wild-type plants depends on the degree of adaptation of the viral isolate to *A. thaliana*.”

Line 236 - typo should read: "and 21-fold"

Changed.